# Are political-opinion pollsters missing ambivalence: "I love Trump"… "I hate Trump"

**James C. Camparo** [1]*, **Lorinda B. Camparo** [2]

**1** Physics and Astronomy Department, Whittier College, Whittier, California, United States of America,
**2** Psychological Sciences Department, Whittier College, Whittier, California, United States of America

* james.c.camparo@aero.org

**Data Availability Statement:** All relevant data are within the manuscript and its Supporting Information files.

**Funding:** The authors received no specific funding for this work.

## Abstract

Given the increasing attention ambivalence is receiving from the psychological community, it must be asked if pollsters' (routinely) dichotomous political opinion surveys are missing something crucial. To determine if there is any legitimacy to this question, undergraduates attending a Liberal Arts college in Southern California were asked to rate their level of agreement/disagreement to 28 statements regarding President Trump in two studies, with the items drawn from actual Quinnipiac (Q) and Brookings Institute (BI) surveys. To quantify ambivalence participants were told they could mark one <u>or</u> two responses per item, with double-responses serving as a measure of ambivalence. In Study 1, mean Trump approval ratings divided along party lines, and were consistent with the Q and BI findings. Nonetheless, approximately 40% of participants registered some level of ambivalence across all political-party affiliations, with those defining themselves as Neither Democrats (DEMs) nor Republicans (REPs) showing the greatest degree of ambivalence. In Study 2, ambivalence towards President Trump was examined looking at both party affiliation and political ideology (Conservative, Moderate, and Liberal). Again, roughly 40% of participants displayed some level of ambivalence, with greater degrees of ambivalence for Independents relative to DEMs and REPs, and Moderates relative to Liberals. Given research indicating that ambivalence is associated with delayed decision making and decisions based on "in the moment" contextual information, our findings our suggestive: if political opinion pollsters do not assess ambivalence, they may be missing information on a fair-sized demographic that could influence an election based on negative information (real or fictitious) surfacing only days before an election… as it did in 2016.

## Introduction

Since the 2016 surprise election of Donald Trump as President of the United States, much has been written regarding the failure of political opinion polls to reliably forecast the potentiality of that outcome. Among a number of conclusions regarding "what happened," researchers have concluded that: (a) though the reliability of *national* political-opinion polling was consistent with historical precedent [1,2], political-opinion polling failed to accurately capture the *electoral college* vote [2,3], which decides presidential election outcomes in the United States;

**Competing interests:** The authors have declared that no competing interests exist.

(b) modeling of likely-voters' opinions was poor [2,4], and importantly for the present work (c) candidate choices were finalized among voters only days before the election [5]. In addition, there is some evidence that social desirability may have played a role in the 2016 polling results [3], though that conclusion is not without argument [2].

An important model for understanding the psychology of political-opinion polling, and thereby explaining the 2016 presidential-election phenomenon, is that of Zaller and Feldman [6]. Based in part on Wilson and Hodges's [7] recognition that the expression of an attitude is affected by an individual's salient thoughts and feelings at the time the attitude is accessed, Zaller and Feldman posited three psychological axioms of political-opinion surveys (noted here in specific reference to a presidential election survey):

(a)  *The ambivalence axiom*–Most people can be expected to have both favorable and unfavorable attitudes towards presidential candidates.

(b)  *The response axiom*–Individuals will respond to survey items by averaging across those candidate-specific attitudes that are most salient at the time.

(c)  *The accessibility axiom*–The accessibility of salient attitudes will depend on both a stochastic sampling of the individual's attitudes, and the political context the individual finds himself/herself in.

Here, we have modified Zaller and Feldman's original accessibility axiom to include the work of Keele and Wolak [8] regarding ambivalence and political context. The important point to note in these axioms is the central role played by ambivalence (*i.e.*, simultaneous favorable and unfavorable attitudes), and that without accounting for ambivalence the interpretation of political-opinion survey results may be incomplete or of no predictive value.

Understanding the subtleties of political-opinion polling is important, because these polls can winnow the debate stage of candidates, affecting voter choices; they affect candidates' campaign strategies, and thereby voter knowledge of policy positions and candidate demeanor; and they can affect voter turnout [9]. Given that professional political-opinion polling as presently conducted fails to concretely take ambivalence into account, the present work wanted to address the question of whether or not this is an issue of any legitimate concern.

## Ambivalence and its measure

Attitudinal ambivalence can be defined as "the simultaneous occurrence of positive and negative implicit or explicit evaluations of a single attitude object" [10], and would appear to be more prevalent in individuals' political opinions than is often recognized [11,12]. In particular, researchers have found that ambivalence tends to be greater among those who engage in effortful cognitive processing [13], that ambivalence is a dynamic process, with presidential candidate ambivalence generally decreasing over the course of a political campaign [14], and that ambivalence disturbs the correlation between candidate vote-intention and actual candidate vote-choice [15].

Though clearly important, quantifying ambivalence can be problematic. One can, of course, question individuals regarding their subjective (*i.e.*, "felt") ambivalence, but transitioning from those self-reports to an objective interval-scale measure is difficult. Most objective measures of ambivalence trace back to Kaplan [16], where valence assessments are made of an attitude object providing a "favorable score" and an "unfavorable score." These two scores are then combined using one of several mathematical formulae, yielding an interval-scale assessment of the respondent's ambivalence [17–19]. These may be defined as V-measures of ambivalence, V for valence [20].

Unfortunately, V-measures have limitations. Most importantly, what one actually quantifies with V-measures are oscillations in attitude, or what may be termed *vacillating* ambivalence. With regard to presidential polling, vacillating ambivalence corresponds to temporal fluctuations between favor/disfavor evaluations of a candidate as the individual responds to survey items. Vacillating ambivalence, however, should be distinguished from *simultaneous* ambivalence, which can be thought of as "in-the-moment," concurrent favor/disfavor evaluations of the presidential candidate; and Camparo and Camparo [20] have obtained evidence suggesting that vacillating and simultaneous ambivalence are two distinct constructs.

To tap into simultaneous ambivalence, recent work has focused on allowing individuals to "double-respond" to questionnaire items [20,21]. For example, a researcher might ask an individual to respond to the statement "President Trump has created a strong economy" on a 5-point ordinal scale ranging from Strongly Disagree to Strongly Agree. As is normally the case, the individual can choose to respond using one response category (*e.g.*, Agree), or they can choose to respond with two response categories (*e.g.*, Agree and Disagree). Clearly, the choice to respond with two response categories on opposite or widely spaced points of the scale is an indication of simultaneous ambivalence. When referring to ambivalence in what follows we will be referring to only this form of ambivalence.

The method of analyzing double-responses to questionnaire items is based on the density matrix [21], which is an array-based tool for tabulating single and double responses. This tool is discussed more fully in Appendix A, but in brief it counts a respondent's number of Strongly Disagree, Disagree, *etc.* responses, whether these occur singly or doubly for a given questionnaire item. From those counts, the array can be employed to compute (for example) an individual's approval score, $A_P$, for President Trump. Additionally, it can be employed to yield a respondent's probability to double-respond to a questionnaire item, which we refer to as the individual's degree of ambivalence, D.

In the present two studies, we constructed ordinal-response items from 2017 and 2018 Quinnipiac (Q) and Brookings Institute (BI) surveys regarding the presidency of Donald Trump, closely following the wording used by those surveys during their data collection. Though Donald Trump is arguably a polarizing political figure, we nevertheless hypothesized that ambivalence would exist in response to these professional political-opinion survey items, which were clearly constructed to elicit dichotomous responses. Verification of the hypothesis would suggest that even professionally designed surveys miss ambivalence in their assessment of political attitudes. Further, we hypothesized that individuals identifying as "Neither Democrat nor Republican" (NRD) would display relatively higher degrees of ambivalence given their lower "loyalty" to the Republican and Democrat parties.

## Study 1

### Method

The first study was conducted in two waves: One in the winter of 2017 and the other in the spring of 2018 (with research approved by the Whittier College Institutional Review Board–approval number 2018F0005), and our goals were twofold. Given that our sample population was drawn from College undergraduates, and thereby represents an important emerging cohort of voters in the general population, we nonetheless recognized that it is a restricted population. Consequently, we first wanted to assure ourselves that surveying these participants would generally reproduce the results from the Brookings Institute and Quinnipiac surveys, which were conducted nationally. To be clear, it is not our intent in the present studies to argue that our findings must generalize to the population at large. Rather, if our sample population produces results consistent with the Q and BI surveys, and also uncovers reasonably

significant levels of ambivalence using the Q and BI items, then it would be fair to conclude that significant levels of ambivalence likely exist in the wider voting population and are missed by professionally designed survey items. Additionally, we wanted to examine the hypothesis that NRD individuals would show greater ambivalence towards Donald Trump's presidency than would Republicans or Democrats. Again, while caution should be exercised generalizing results from our restricted sample population to voters at large, a positive result would none-theless motivate analyses of more representative general voting populations.

## Materials

A 28-item questionnaire was developed taking items from an April 2017 Brookings Institute survey (five items) and an August 2017 Quinnipiac survey (23 items). The items were presented to participants as Agree/Disagree statements on a 7-point ordinal scale regarding Trump's presidency, with responses ranging from Very Strongly Disagree (Score = −3) to Very Strongly Agree (Score = +3). The wording of the items was kept as close to the original as possible, and only changed to make sense when presented as a 7-point Agree/Disagree statement. Two surveys were created, with positive statements in one reworded to negative statements in the other in order to guard against potential response biases. Since the Brookings Institute and Quinnipiac responses to the items were available and parsed by party affiliation, it was possible to compare our survey's Trump-approval scores to those derived from the Brookings Institute and Quinnipiac national surveys. The questionnaire items for Study #1 are provided in Appendix B.

## Participants and procedure

Participants were undergraduates (N = 105; 75 female) from a small Liberal Arts college in Southern California and were recruited via a brief explanation of the project by one of the authors or a research assistant during the participants' class time. Students were told they would be asked to complete a questionnaire about their views on social issues at a pre-arranged time and place on campus. Additionally, students were told they might receive extra credit for participation, but that the decision would be up to the instructor of the class they were recruited from. All procedures adhered to Institutional Review Board (IRB) guidelines.

Students who agreed to participate met at the pre-arranged location at the appropriate time and were given a packet composed of an informed consent form, a demographics questionnaire, and the 28-item questionnaire regarding Trump's presidency. After participants completed the forms and questionnaire, they were debriefed. As part of the demographics, participants reported their political-party affiliations: Republican (REP), Democrat (DEM), Independent (IND), or Other (OTH). To increase power, we combined Independents and Other into a "Neither Republican nor Democrat" category (NRD): $N_{REP} = 13$, $N_{DEM} = 56$, $N_{IND} = 19$, $N_{OTH} = 15$.

Prior to filling out the questionnaire, participants were explicitly told that they could mark one or two responses for any item. Double-responding (*i.e.*, simultaneous ambivalence) was only coded as such if the double responses were not adjacent to each other on the 7-point ordinal scale (*e.g.*, 1 and 3; 2 and 4, 4 and 6). If the double responses were adjacent to each other, a coin was flipped to decide between the two responses. Our purpose with this criterion for double-responding was to distinguish those participants truly indicating ambivalence from those who might have simply wanted a finer ordinal-scale division.

Data were entered into a density matrix array for each participant. Using the diagonal elements, we computed the participant's President Trump approval score (labeled with k,

indicating the participant's ID), $A_P(k)$, which ranged from $-3$ to $+3$ (negative to positive approval), and which could be compared with the Brookings Institute and Quinnipiac survey results.

Previous work [20] has shown that the probability distribution for simultaneous ambivalence follows a mixture model:

$$P[M] = [(1 - F) + Fe^{-(1/L)}][1 - H(M)] + FH(M)e^{-(1/L)}\left(\frac{1}{L^M M!}\right). \qquad (1)$$

Here, P[M] is the probability for obtaining M double responses on a survey; F is the fraction of a population manifesting ambivalence (*i.e.*, that fraction of a population that double responds); L is the average number of double-responses per participant in the ambivalent population, and H[M] is the Heaviside Unit-Step function (*i.e.*, H[0] = 0, and H[M] = 1 for $M \geq 1$). The values for F and L were determined via Bayesian analysis using Markov-Chain Monte Carlo [22]. Dividing L by the total number of questionnaire items, N, gives the probability that an ambivalent respondent will double-respond to an item; we define this as the degree of ambivalence, D = L/N. F and D were our primary measures of ambivalence.

## Results

Since the diagonal elements of the density matrix provide probabilities for approval and disapproval of Donald Trump's presidency, it is possible to compute mean approval/disapproval scores for each participant from the density matrix, and then average these to obtain mean values and standard deviations of Trump approval for the various party affiliations:

$$A_P(k) = \sum_{J=1}^{7} (J - 4)\rho_{JJ}(k). \qquad (2)$$

Here, the $\rho_{JJ}(k)$ are the diagonal elements of the 7x7 density matrix for each participant k, and (J−4) corresponds to the interval-scale number assigned to each ordinal category. Our results compared well to the Brookings Institute and Quinnipiac results as shown in Fig 1 and collected in Table 1. Error bars in the figure correspond to 95% confidence intervals (CIs) of the mean [23]. The CI is relatively large for Republicans, due to their small number in our sample. Though Brookings Institute and Quinnipiac only reported results for Republicans, Independents, and Democrats, our larger group of Neither Republican nor Democrat displayed Trump-approval scores very similar to Independents. Overall, our mean $A_P$ values are very much in line with the Brookings Institute and Quinnipiac professional results.

Fig 2A and 2B show our results related to ambivalence, with the values collected in Table 2. Fig 2A shows the fraction of the population exhibiting any level of ambivalence, F, while Fig 2B shows the degree of ambivalence, D. Ambivalence was primarily manifested as Agree/Disagree double responses rather than (for example) Strongly-Agree/Strongly-Disagree double responses. Again, 95% CIs for Republicans are large due to their small number in our sample. Though there were no significant differences in F across political affiliation, we do see that across all political affiliations ambivalence was present at roughly the 30% to 50% level, and significantly different from zero. Further, regarding Table 2 the degree of ambivalence for NRDs was approximately 0.08. Though this indicates that on average ambivalent NRD respondents only gave two double-responses across the entire 28-item questionnaire, it must be remembered that these professional survey items were designed to elicit *binary* choices and not probe for (or elicit) ambivalence.

The confidence intervals in Fig 2B suggest that there is a difference in the degrees of ambivalence between NRDs and DEMs [23]. This is demonstrated rigorously in Fig 3, which shows

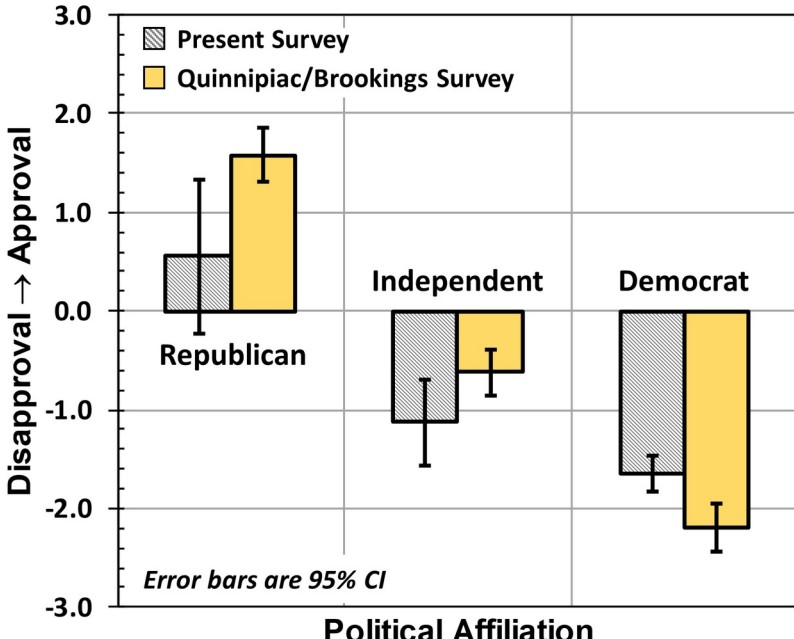

**Fig 1. Study-1 results regarding approval of Donald Trump's presidency, which are consistent with the Brookings Institute and Quinnipiac results.**

party-affiliation differences in D, computed by differencing the $10^6$ Markov Chain Monte Carlo results for L (obtained for the three party affiliations) and then dividing by N. At the 95% confidence level, NRDs had higher levels of ambivalence than DEMs, and the Cohen's d effect size for the difference was 0.69 indicating a medium-to-large effect [24]. Taken together these results partially support our Study 1 hypotheses: Trump approval scores are consistent with those obtained by Quinnipiac and Brookings Institute, and NRDs show a larger degree of ambivalence than Democrats at a statistically significant level.

## Study 2

### Method

The second survey study was again conducted in two waves: One in the fall of 2018 and the other in the spring of 2019 (with research approved by the Whittier College Institutional Review Board–approval number 2019S0010). In this study, we wanted to consider ambivalence towards Trump's presidency as it relates to political ideology as well as party affiliation. Though political ideology is certainly linked to party affiliation, Barber and Pope [25] have shown that party identification can be much stronger than ideology in terms of attitude

**Table 1. Trump approval score $A_P$ broken down by party affiliation.**

| Affiliation | Present Study | | Brookings Institute & Quinnipiac | |
|---|---|---|---|---|
| | Mean | 95% CI | Mean | 95%CI |
| Republican | 0.56 | [-0.22,1.34] | 1.58 | [1.31,1.85] |
| Independent | -1.13 | [-1.57,-0.69] | -0.62 | [-0.85,-0.39] |
| Neither REP nor DEM | -1.12 | [-1.44,-0.80] | | |
| Democrat | -1.65 | [-1.83,-1.47] | -2.19 | [-2.44,-1.94] |

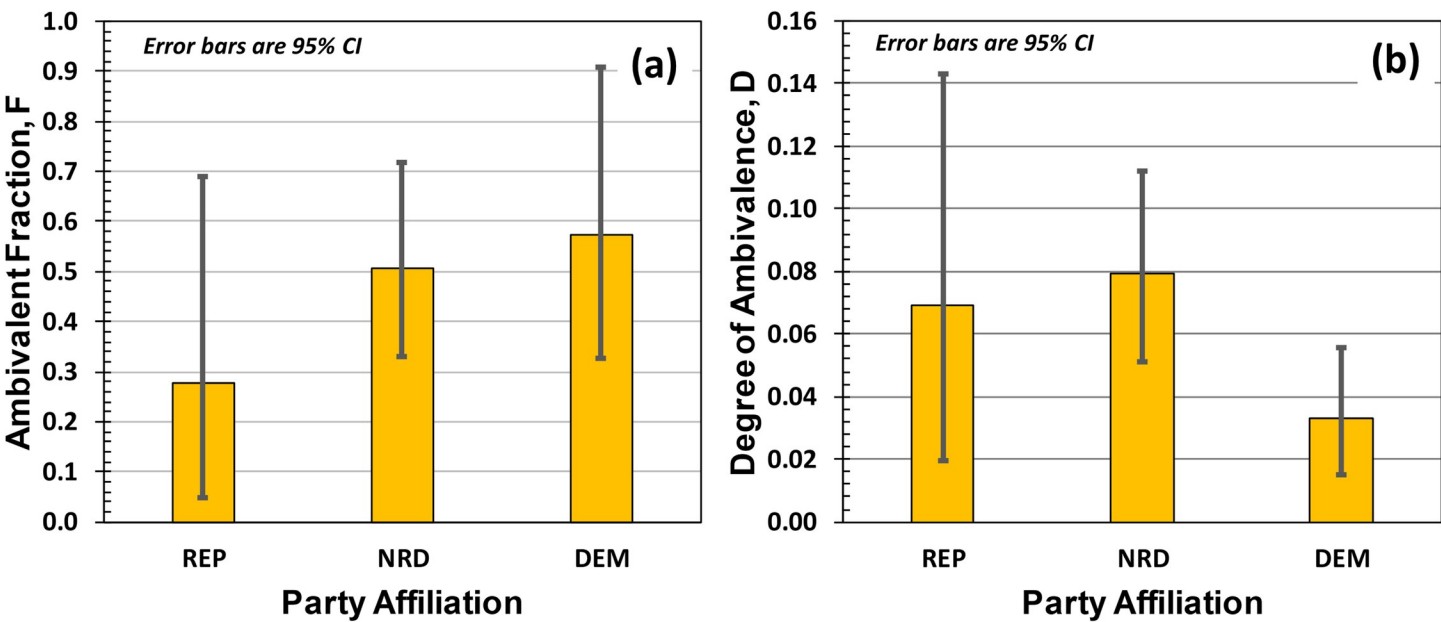

**Fig 2.** (a) Fraction of the population showing ambivalence, F, which in all cases looks to be around 30% to 50%. (b) Degree of ambivalence, D, which is the probability that an ambivalent participant will double-respond to a questionnaire item.

expression. Further, during the 2020 presidential election it became clear that Democrats as a party were composed of ideologically distinct moderates and progressives, whereas Republicans as a party were composed of ideologically distinct ardent Trump supporters and traditional conservatives. Given these empirical results, we felt it important to explore potential ambivalence differences related to political ideology in addition to party affiliation.

We also wanted to examine those aspects of Trump's presidency that might be most conducive to arousing ambivalence: (a) an overarching or "general" aspect (*e.g.*, "I approve of the way Donald Trump is handling his job as president"), (b) foreign policy, (c) the economy, (d) immigration, (e) the Mueller probe, and (f) interactions with the media. We hypothesized that: (a) Trump-approval scores would follow an ascending pattern of DEM → IND → REP and Liberal (LIB) → Moderate (MOD) → Conservative (CON); (b) Independents/Moderates would show greater degrees of ambivalence than Democrats/Republicans and Liberals/Conservatives; (c) that political ideology results for Trump approval and ambivalence would mirror the party affiliation results (*e.g.*, that IND ambivalence would be indistinguishable from MOD ambivalence, and that REP Trump-approval would be indistinguishable from CON

**Table 2. Study-1 ambivalence parameters.** The degree of ambivalence D only refers to that fraction of the population that showed any level of ambivalence.

| Affiliation | Fraction of Population Identified as Ambivalent, F | | Degree of Ambivalence, D | |
|---|---|---|---|---|
| | Mean | 95% CI | Mean | 95%CI |
| Republican | 0.28 | [0.05, 0.69] | 0.07 | [0.02, 0.14] |
| Neither REP nor DEM | 0.51 | [0.33, 0.72] | 0.08 | [0.05, 0.11] |
| Democrat | 0.57 | [0.33, 0.91] | 0.03 | [0.02, 0.06] |

D is the probability that an ambivalent participant will double-respond to an arbitrary questionnaire item.

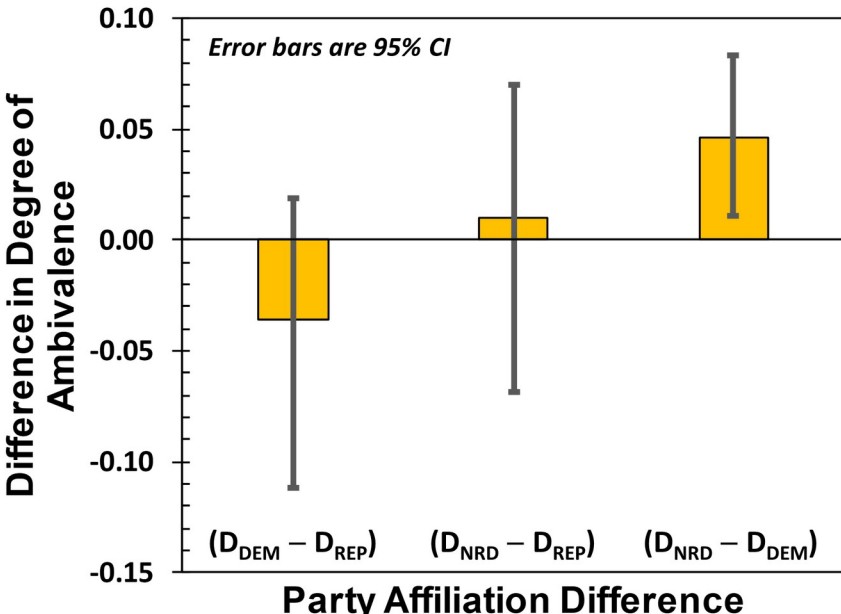

**Fig 3. Difference in degrees of ambivalence, D.** At the 95% confidence level, those respondents identifying as Neither Republican nor Democrat (NRD) have higher degrees of ambivalence than Democrats.

Trump-approval), and (d) that some aspects of Trump's presidency would be more conducive to arousing ambivalence in Independents and Moderates than other aspects.

## Materials

A 28-item questionnaire was again developed, this time taking items solely from a number of Quinnipiac surveys. All but one of the items were drawn from Quinnipiac surveys between June and August 2018, with one of the items (Item 26 related to America's relations with its European allies) drawn from a June 2017 Quinnipiac survey. The items were presented to participants as Agree/Disagree statements on a 5-point ordinal scale this time ranging from Strongly Disagree (Score = −2) to Strongly Agree (Score = +2). (Originally, we had hoped that providing respondents with a 7-point ordinal scale would provide subtler distinctions in ambivalence; since this was not the case in Study 1, we reverted to a 5-point ordinal scale in Study 2.) The wording of the items was again kept as close to the original as possible and only changed to make sense when presented as a 5-point Agree/Disagree statement. Again, two surveys were created, with positive statements in one reworded to negative statements in the other in order to guard against potential response biases. Unfortunately, for Study 2 the individual Quinnipiac item responses were not readily available, and so it was not possible to compare our Trump-approval results to Quinnipiac's national survey responses. The questionnaire items for Study 2 are provided in Appendix C.

## Participants and procedure

Participants were undergraduates (N = 153; 87 female) from a small Liberal Arts college in Southern California, and were recruited via a brief explanation of the project by one of the authors or a research assistant during the participants' class time. Students were told they would be asked to complete a questionnaire at a pre-arranged time and place on campus. Additionally, students were told they might receive extra credit for participation, but that the

decision would be up to the instructor of the class they were recruited from. All procedures adhered to IRB guidelines.

Students who agreed to participate met at the pre-arranged location at the appropriate time, and were given a packet composed of an informed consent form, a demographics questionnaire, and the 28-item questionnaire regarding Trump's presidency. After participants completed the forms and questionnaire, they were debriefed. As part of the demographics, participants reported their political party affiliations as Republican (REP), Democrat (DEM), or Independent (IND): $N_{REP} = 20$, $N_{DEM} = 83$, $N_{IND} = 40$. (Given the larger number of participants reporting their party affiliation as Independent, we found it unnecessary to create an NRD category.) Additionally, participants self-reported their political ideology as Conservative (CON), Liberal (LIB), or Moderate (MOD): $N_{CON} = 22$, $N_{LIB} = 55$, and $N_{MOD} = 73$.

Prior to filling out the questionnaire, participants were explicitly told that they could mark one <u>or</u> two responses for any item. Again, double-responding was only coded as such if the double responses were not adjacent to each other on the 5-point ordinal scale (*i.e.*, 1 and 3, 2 and 4, 3 and 5). If the double responses were adjacent to each other, a coin was flipped to decide between the two responses. Our purpose with this criterion for double-responding was to distinguish those participants truly indicating ambivalence from those who might have simply wanted a finer ordinal-scale division.

Similar to Study 1, data were entered into a density matrix array for each participant. Using the diagonal elements, we computed a Trump approval score for each participant, $A_P(k)$, which ranged from −2 to +2 (negative to positive approval). Additionally, we performed Bayesian analysis using Markov-Chain Monte Carlo to determine mean values and 95% CI intervals for F (the fraction of the population exhibiting ambivalence) and L (the number of double responses for that fraction of the population identified as ambivalent). In addition to looking at all 28-item responses collectively, each of our six Aspect-of-Trump's-Presidency categories were analyzed separately, so that $L/N_{cat}$ (*i.e.*, the number of double responses for the category divided by the total number of items for the category, $N_{cat}$) yielded that category's degree of ambivalence for the participant.

## Results

Similar to Study 1, we used the diagonal elements of the density matrix to compute Trump approval scores for each participant, $A_P(k)$, now however scaled from −2 to +2. We then averaged these individual scores to obtain mean values and standard deviations of Trump approval for party affiliation and political ideology:

$$A_P(k) = \sum_{J=1}^{5} (J - 3)\rho_{JJ}(k) \tag{3}$$

Here, the $\rho_{JJ}(k)$ are the diagonal elements of the 5x5 density matrix for each participant k, and (J−3) corresponds to the interval-scale number assigned to each ordinal category. As illustrated in Fig 4, REPs and CONs showed the highest levels of approval, INDs and MODs intermediate/low levels of approval, and DEMs and LIBs displayed the lowest levels of approval. The numerical values are collected in Table 3.

Consistent with our hypotheses, mean Trump approval scores differed significantly for party affiliation as determined by one-way ANOVA: $F(2,52.0) = 44.0$, $p < 0.001$. (Due to a violation of equality of variances, we employed the Brown-Forsythe homogeneity correction; similar results were obtained with the Welch homogeneity correction.) Post-hoc t-tests revealed that each party-affiliation's approval score was statistically different from the other two party-affiliation approval scores at $p < 0.05$. Considering the three party-affiliation groups, the between-

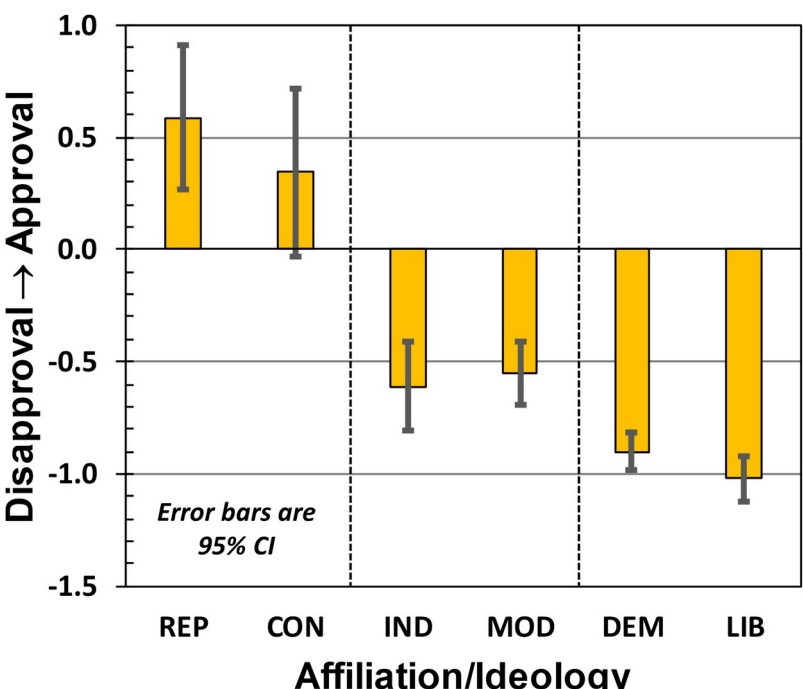

**Fig 4. Study-2 results regarding approval of Donald Trump's presidency.**

group standard deviation was 0.79 (*i.e.*, we computed the standard deviation among the three groups' mean Trump-approval scores: $M_{REP} = +0.59$, $M_{IND} = -0.61$, and $M_{DEM} = -0.90$).

Similarly, mean Trump-approval scores for political ideology differed significantly as determined by one-way ANOVA: $F(2,42.7) = 33.3$, $p < 0.001$. (Again, due to a violation of equality of variances, we employed the Brown-Forsythe homogeneity correction; similar results were obtained with the Welch homogeneity correction.) Post-hoc t-tests revealed that each political-ideology's approval score was statistically different from the other two political-ideology approval scores at $p < 0.001$. Considering the three political-ideology groups, the between-group standard deviation was 0.69 (*i.e.*, we computed the standard deviation among the three groups' mean Trump-approval scores: $M_{CON} = +0.34$, $M_{MOD} = -0.55$, and $M_{LIB} = -1.02$).

Contrary to our hypotheses, we did find a significant difference in Trump-approval scores between DEMs (M = −0.90, SD = 0.39) and LIBs (M = −1.02, SD = 0.37) using a one-tailed t-test: t (136) = 1.77, p = 0.04. However, considering the between-group standard deviations for party affiliation and political ideology as a normative parameter (*i.e.*, their pooled between-group variance), the effect size of this DEM/LIB approval score difference is only 0.16, which is a small effect [24]. Other party/ideology differences were not significant (except for obvious ones like REP/MOD).

Fig 5 provides a more nuanced understanding of the approval scores, where approval is considered as a function of various aspects of Trump's presidency. Broadly speaking, INDs and DEMS (as well as MODs and LIBs) have varying levels of Trump approval based on the

**Table 3. Trump approval score $A_P$ broken down by party affiliation and political ideology.**

|        | REP         | CON          | IND           | MOD           | DEM           | LIB           |
|--------|-------------|--------------|---------------|---------------|---------------|---------------|
| **MEAN**   | 0.59    | 0.34         | −0.61         | −0.55         | −0.90         | −1.02         |
| **95% CI** | [0.27,0.91] | [−0.03,0.72] | [−0.81,−0.41] | [−0.69,−0.41] | [−0.99,−0.81] | [−1.12,−0.92] |

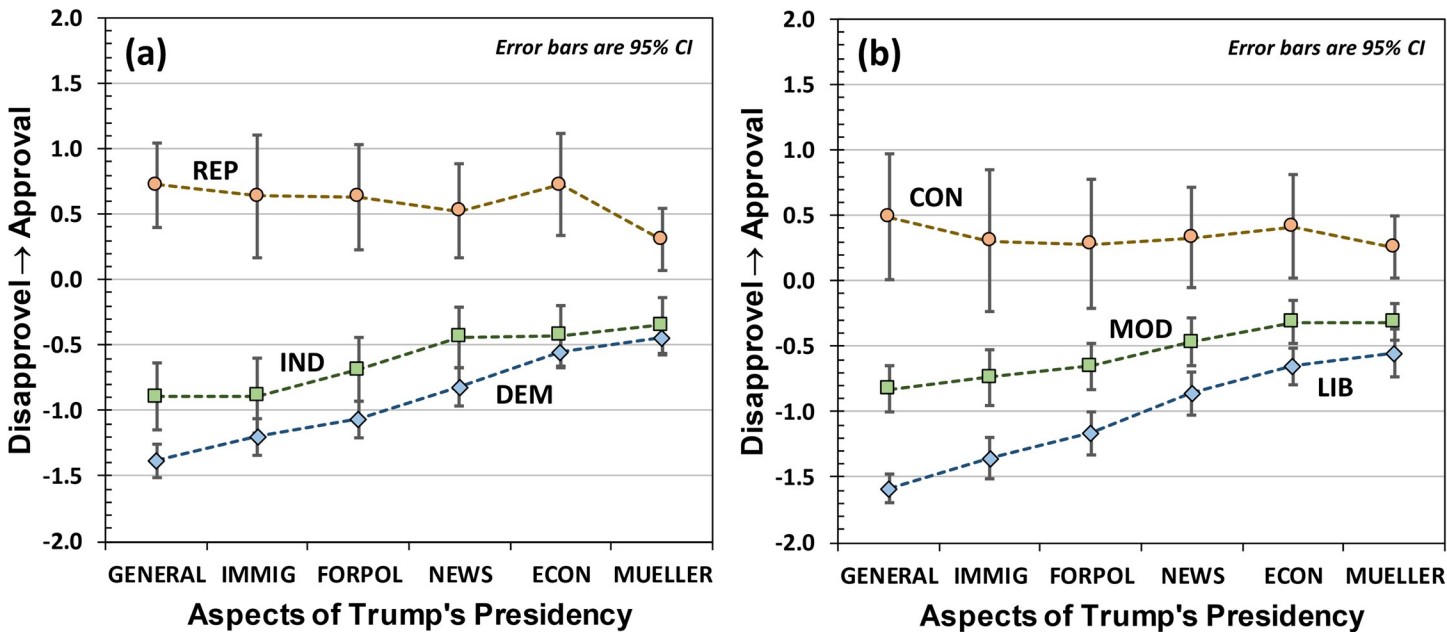

**Fig 5. Approval score for various aspects of Trump's presidency: GENERAL aspects of the presidency, immigration (IMMIG), Foreign Policy (FORPOL), interactions with the news media (NEWS), the economy (ECON), and the Mueller probe (MUELLER).** (a) is for Republicans (REP), Independents (IND), and Democrats, (b) is for Conservatives (CON), Moderates (MOD), and Liberals (LIB). Error bars are again 95% confidence intervals.

aspect of Trump's presidency under consideration. Interestingly, when asked to consider Trump's presidency generally, INDs and DEMs (as well as MODs and LIBs) have greater levels of Trump disapproval than when asked to rate almost any other *specific* aspect of Trump's presidency. These conclusions are validated by repeated measures ANOVA using Aspect-of-Trump's-Presidency as the within-participant variable: for INDs $F(5,195) = 10.21$ with $p < 0.001$, for DEMs $F(5,410) = 45.15$ with $p < 0.001$, for MODs $F(5,360) = 14.02$ with $p < 0.001$, and for LIBs $F(5,270) = 39.04$ with $p < 0.001$. For REPs and CONs, the repeated measures ANOVAs indicated that there were no statistically significant differences among Aspects-of-Trump's-Presidency categories at $p < 0.05$ levels.

Fig 6A shows the fraction of ambivalent participants as a function of party affiliation and political ideology, while Fig 6B shows the degree of ambivalence (see also Table 4). There is no statistically significant difference in the fraction of ambivalent participants with party affiliation or political ideology; again 30% to 50% of respondents displayed some level of ambivalence. However, INDs showed a significantly higher degree of ambivalence than either REPs or DEMs, which is consistent with our hypotheses and the results of Study 1. It is worth noting that the effect is so strong for party affiliation that it is evidenced simply by an examination of bar heights and 95% CI intervals: the 95% CIs do not overlap [23]. There is no statistically significant difference in degree of ambivalence between REPs and DEMs. Regarding political ideology, there is no significant difference in the degree of ambivalence between MODs and CONS, whereas there is a statistically significant difference between MODs and LIBs ($M_{MOD} - M_{LIB} = 0.047$, 95% CI = [0.008,0.084]). However, compared to party affiliation the difference in degree of ambivalence between MODs and LIBs is smaller than that between INDs and DEMs.

To look for ambivalence variability between party-affiliation and the corresponding political ideology (*e.g.*, REP affiliation and CON ideology), we differenced the $10^6$ Markov Chain Monte Carlo results for L (obtained separately for party affiliation and political ideology) and then divided these $10^6$ differences by N. The variation in degree of ambivalence is shown in

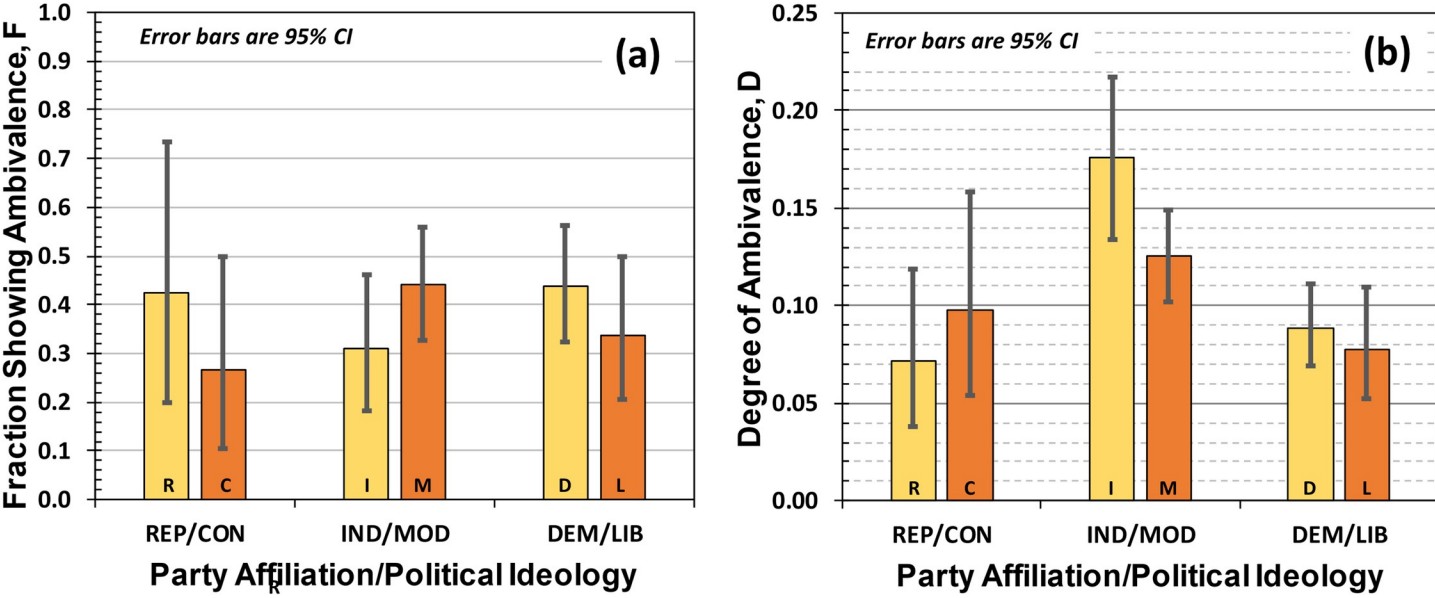

**Fig 6.** (a) Fraction of the population showing ambivalence, F, which in all cases looks to be around 30% to 40%. (b) Degree of ambivalence, D, which is the probability that an ambivalent participant will double-respond to a questionnaire item.

Fig 7 (*e.g.*, D for REPs minus D for CONs). Interestingly, while we see no significant difference in the degree of ambivalence between REPs/CONs on the one hand, and DEMs/LIBs on the other, there is a statistically significant difference between INDs/MODs (($M_{IND} - M_{MOD} = 0.051$ with 95% CI = [0.001,0.105])) with those identifying as Independent having higher levels of ambivalence than those who view their ideology as Moderate.

Fig 8 shows the degree of ambivalence for differing Aspects-of-Trump's-Presidency grouped according to party affiliation (Fig 8A) and political ideology (Fig 8B). Since degree of ambivalence can be computed for each participant individually, based on their individual density matrix, it is possible to perform a repeated measures ANOVA in the case of D. Contrary to expectations, repeated measures ANOVA on D showed no significant difference among Aspects-of-Trump's-Presidency categories for party affiliation or political ideology. In part, this is due to low power, which is a combination of the 30–50% fraction of the population that shows ambivalence and the fact that each aspect of Trump's presidency was probed with only four or five items. Since the fraction of ambivalence is a population variable, it is not possible to perform a repeated measures ANOVA considering F.

## Discussion

In both Study 1 and Study 2, we found that roughly 40% of respondents indicated some level of ambivalence, across all party affiliations and political ideologies; and while it might be

**Table 4. Ambivalence parameters F and D broken down by party affiliation and political ideology.**

|  | REP | CON | IND | MOD | DEM | LIB |
|---|---|---|---|---|---|---|
| **F: MEAN** | 0.42 | 0.27 | 0.31 | 0.44 | 0.44 | 0.34 |
| **F: 95% CI** | [0.20,0.73] | [0.11,0.50] | [0.18,0.46] | [0.33,0.56] | [0.32,0.56] | [0.21,0.50] |
| **D: MEAN** | 0.07 | 0.10 | 0.18 | 0.13 | 0.09 | 0.08 |
| **D: 95% CI** | [0.04,0.12] | [0.05,0.16] | [0.13,0.22] | [0.10,0.15] | [0.07,0.11] | [0.05,0.11] |

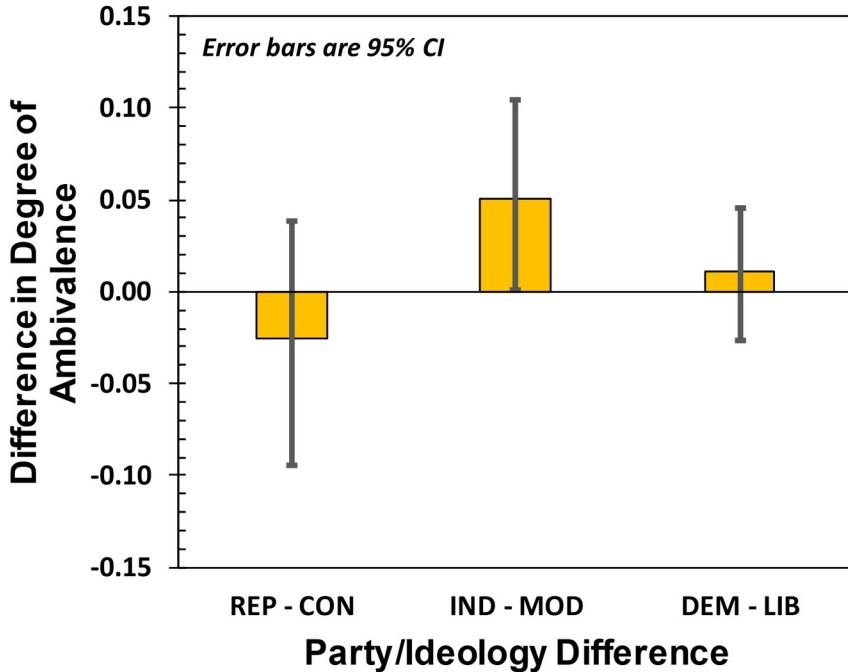

**Fig 7. Difference in degree of ambivalence, D between party affiliation and the corresponding political ideology.**

argued that the degree of ambivalence was relatively small (*i.e.*, roughly 0.1 on a zero to 1.0 scale), it must be remembered that this degree of ambivalence was seen through the lens of

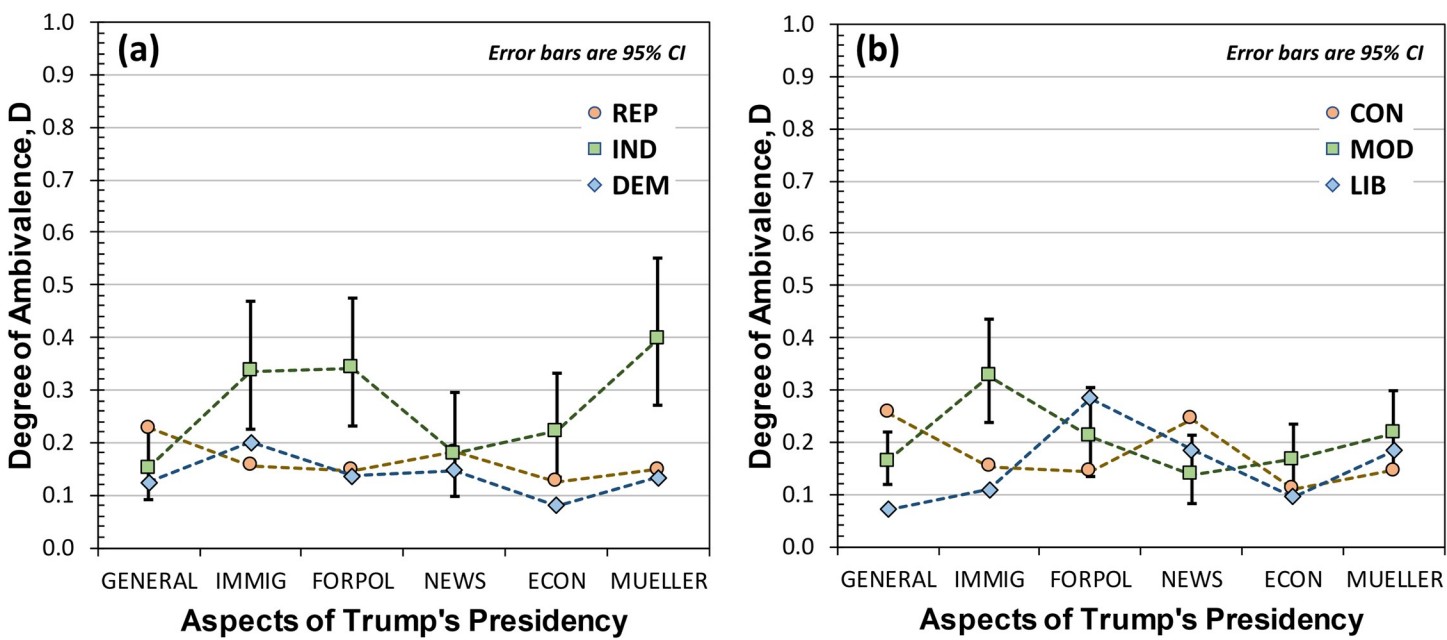

**Fig 8. Degrees of ambivalence for various aspects of Trump's presidency: GENERAL aspects of the presidency, immigration (IMMIG), Foreign Policy (FORPOL), interactions with the news media (NEWS), the economy (ECON), and the Mueller probe (MUELLER).** (a) is for *Republicans* (REP), *Independents* (IND), and *Democrats*, (b) is for *Conservatives* (CON), *Moderates* (MOD), and *Liberals* (LIB). Median values and 95% CIs were determined by Markov Chain Monte Carlo for the probability distribution of Eq (10). For clarity in the graph, 95% CI error bars are only shown for INDs and MODs. Error bars for DEMs (LIBs) were somewhat smaller, while error bars for REPs (CONs) were larger.

professionally developed survey items generally designed to elicit dichotomous responses. Consequently, while the present sample population was restricted to college-age voters (an important but nevertheless sub-set of voters) the results do provide evidence that professional political-opinion surveys are missing important levels of ambivalence in the American electorate. More specifically, while the present results were focused on ambivalence regarding Trump's presidency, it is not much of a stretch to conclude that the general nature of the results likely extends to the broader host of issues political-opinion surveys attempt to assess (*e.g.*, affirmative action and pro-choice/pro-life).

In our studies, the degree of ambivalence for Independents was roughly double that of Republicans and Democrats. When this result is married to the fact that 40% of the electorate identifies as Independent [26], it suggests (recognizing the limitations of our study) that 16% of all voters could have some level of ambivalence and will therefore be sensitive to political context at the time that they make their final voter choice (*i.e.*, 40% of voters are Independents, and 40% of these manifest ambivalence). In close elections, where ambivalent voters wait to cast their ballot on election day, this could easily play a significant role in an election's outcome, and indicates considerable importance to candidate messaging in the days just prior to an election.

To be clear, the ambivalence uncovered here is not ambivalence of Independents as a *group*. It is well known that roughly 46% of Independents lean Democratic, 35% of Independents lean Republican, and that only 19% have no leaning [27]. Thus, as a group Independents likely have ambivalence towards Donald Trump's presidency. However, the ambivalence reported here is at the individual level. Thus, our results suggest that those Independents leaning Democratic or leaning Republican show greater levels of ambivalence at the individual level than those individuals who declare themselves Democrats or Republicans.

It is also necessary to recognize that ambivalence is not equivalent to a centrist perspective or moderation. Regarding the Trump approval scores broken down by Aspect-of-Trump's-Presidency in Fig 5, it would be very easy to walk away from the data with the sense that Independents are looking for "centrist" political positions *between* those endorsed by Republicans and Democrats; that is, they are akin to more "moderate-leaning" Republicans or more "moderate-leaning" Democrats. However, the ambivalent nature of Independents' responses suggests that they may be looking for political positions that somehow *merge* the polarized Republican/Democratic positions. While this may seem like a contradiction in terms, it may be that there are latent factors driving the ambivalence that are unrecognized by the political analysts attempting to frame the debate. It may be that the political issues as seen by Independents do not fall neatly into the categories outlined by Republicans and Democrats. Looking for those latent factors could be important for understanding the political opinions (and consequently the political desires) of nearly half the electorate.

One way that latent factors could manifest in ambivalence is if questionnaire items lend themselves to different interpretations. For example, in Study 2 Item 25 states "I think the nation's economy is getting better." On the one hand, a participant could consider this item in the context of the stock market and think to answer Agree; on the other hand, this same participant could consider this item in the context of America's widening wealth-gap and think to answer Disagree. It should be recognized, however, that this interpretive conflict fits the definition of ambivalence: "the simultaneous occurrence of positive and negative implicit or explicit evaluations of a single attitude object" [10], where the attitude object here is the economy *taken broadly*. In this situation the identification of ambivalence can alert the pollster or researcher to probe more deeply for latent factors (*e.g.*, stock market and wealth-gap). To the extent that the pollster can identify well-delineated latent factors through follow-up probing, more detailed information has been gained on the participant's evaluation of the attitude object. Of course, latent factors resolving ambivalence will rarely be so obvious (if present at

all). More often we expect the presence of latent factors to force a fundamental re-appraisal of the attitude object (*e.g.*, the political issue targeted by the questionnaire item). For example, one could easily imagine that latent factors drive ambivalence towards affirmative action, though teasing those latent factors apart could be a Gordian Knot puzzle.

Another potential latent factor could be ambivalence arising from contagion: the influence of social networks on attitudes (see for example [28] and references therein). For example, some individuals could be more susceptible/vulnerable to others' opinions or views, perhaps due to age, personality, or living conditions (*i.e.*, living alone as opposed to living with a partner). Continuing with this speculation, it may be that ambivalent individuals have social networks with close acting peer-nodes comprising both Democrats and Republicans. For example, an individual living in a family with some members identifying as Democrats and others as Republicans may be more likely to develop an ambivalent attitude towards President Trump. Alternatively, it could be that individuals with generable ambivalence tend to form peer networks with others of similar potential. The development of ambivalence could then be a "feedback loop" process, with these individuals stimulating the formation of each other's ambivalent attitudes.

Though we hypothesized that there would be differences in Independents' and Moderates' ambivalence when parsed according to particular aspects of Trump's presidency, the data of Fig 8 show no significant effect. To some degree, this is certainly related to low power: 40% of Independents and Moderates exhibit ambivalence, and of these the rate of double marking (for the present survey items) was only two in 10. Thus, with four to five items per Aspect-of-Trump's-Presidency the likelihood of obtaining an ambivalent response was too low for any individual aspect included in this survey. However, we cannot ignore the possibility that attitude valence (*i.e.*, Trump approval as illustrated in Fig 5) and attitude ambivalence are distinct cognitive processes. To support this possibility, we note that researchers have found that ambivalent individuals likely integrate incongruent attributes of an attitude object prior to making an attitude evaluation [29], and that ambivalent thinking shows differences in brain activation relative to univalent thinking [30]. Thus, it may be that attitude valence parses more finely than attitude ambivalence, differentiating among (in the present study) aspects of Trump's presidency that ambivalence cannot but help to integrate.

Finally, we note that in Study 2 the number of participants identifying as Democrats was larger than those identifying as Liberal, while the number of participants identifying as Moderate was larger than the number identifying as Independent. We explain this result by noting that party affiliation does not correlate strictly with political ideology, which is a conclusion borne out by the 2020 Democratic Convention where both Moderates and Liberals had a significant presence. The fact that Democrats likely include sizeable fractions of Moderates and Liberals would explain the results of Fig 7, where the ambivalence difference between Independents and Moderates was significant at the 95% CI. Specifically, it may be that a fraction of Moderates were actually Democrats, who are known to have highly negative (and therefore not ambivalent) attitudes towards President Trump, whereas Independents would have greater levels of ambivalence as illustrated in Fig 6B.

Returning to where this paper began (*i.e.*, the 2016 presidential election), we offer a novel argument regarding ambivalence's role in Trump's 2016 election win: (a) a large fraction of the electorate identified as NRD [31]; (b) of these voters, a large fraction had some level of ambivalence regarding Donald Trump; (c) these ambivalent voters likely made their final decision just days before the election [5], and, consequently, were likely influenced by the political context at that time [32]; (d) the political context just days before the election was negative for Hillary Clinton (*i.e.*, Comey's re-opening of the Hillary Clinton email controversy). Though we believe that the potential role of ambivalence in the 2016 presidential election is interesting,

and perhaps worthy of further study, regardless of its veracity the "story" serves as a cautionary tale for the importance of ambivalence in political opinion polling.

## Appendix A: Computations with the density matrix

The density matrix (indicated by the Greek letter rho,$\rho$) provides a means of counting single and double responses to questionnaire items. To illustrate, consider one individual responding to N survey items on a five-point ordinal scale. For the first item, the individual indicates "Agree," and since agree corresponds to category 4 on the ordinal scale one full count is placed in the $4^{th}$-row/$4^{th}$-column box of the 5x5 density-matrix array as illustrated in Fig A1. For the next item, this individual both Agrees <u>and</u> Disagrees. Consequently, since "Disagree" corresponds to category 2 (and since we have one count to apply for this single questionnaire item), one half-count is placed in the $2^{nd}$-row/$2^{nd}$-column box, and the other half-count is placed in the $4^{th}$-row/$4^{th}$-column box (adding to the one count already in the box from the previous item) as illustrated in Fig A2. Then, to account for ambivalence one half-count is placed in the $2^{nd}$-row/$4^{th}$-column box, and one half-count in the $4^{th}$-row/$2^{nd}$-column box. (The mathematical formalism of the density matrix only requires that the sum of counts along the diagonal equal the number of items in the survey [21]). After continuing this counting procedure for all of the survey items, and then dividing each box in the array by N, we obtain the elements of the density matrix.

The diagonal elements of the density matrix give the probabilities for overall agreement with the survey items, overall disagreement, overall strong disagreement, *etc.* For example, if the items are looking at approval of Donald Trump's presidency, then the number in the $4^{th}$-row/$4^{th}$-column box gives the probability that this individual approves of Donald Trump's presidency (*i.e.*, agrees with positive statements); the number in the $5^{th}$-row/$5^{th}$-column

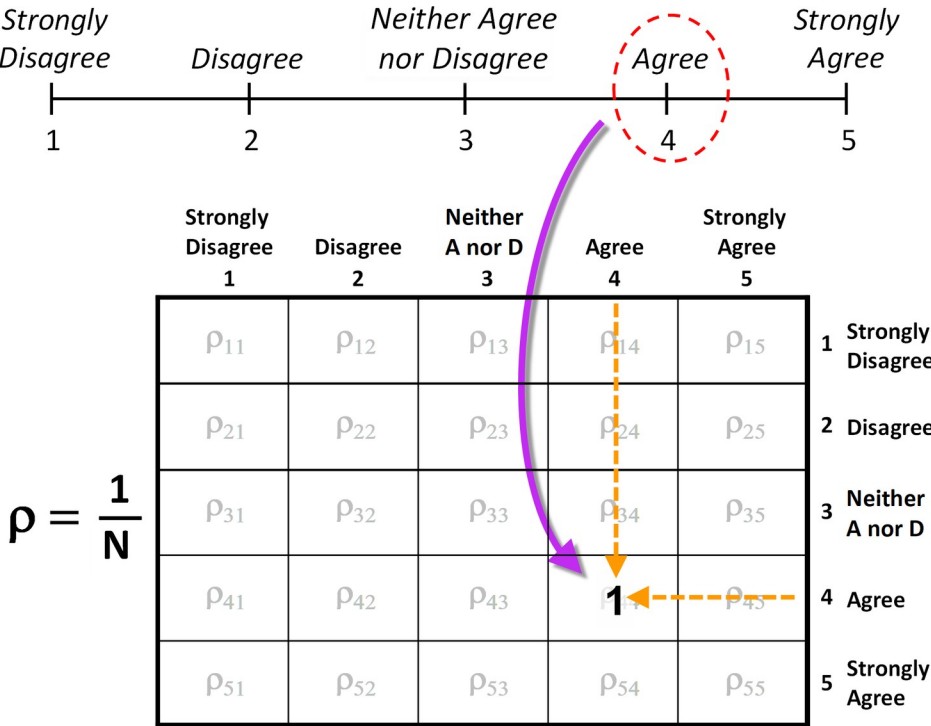

**Fig A1. For a single response to a survey item, one full count is placed in the corresponding diagonal box of the square density-matrix array.**

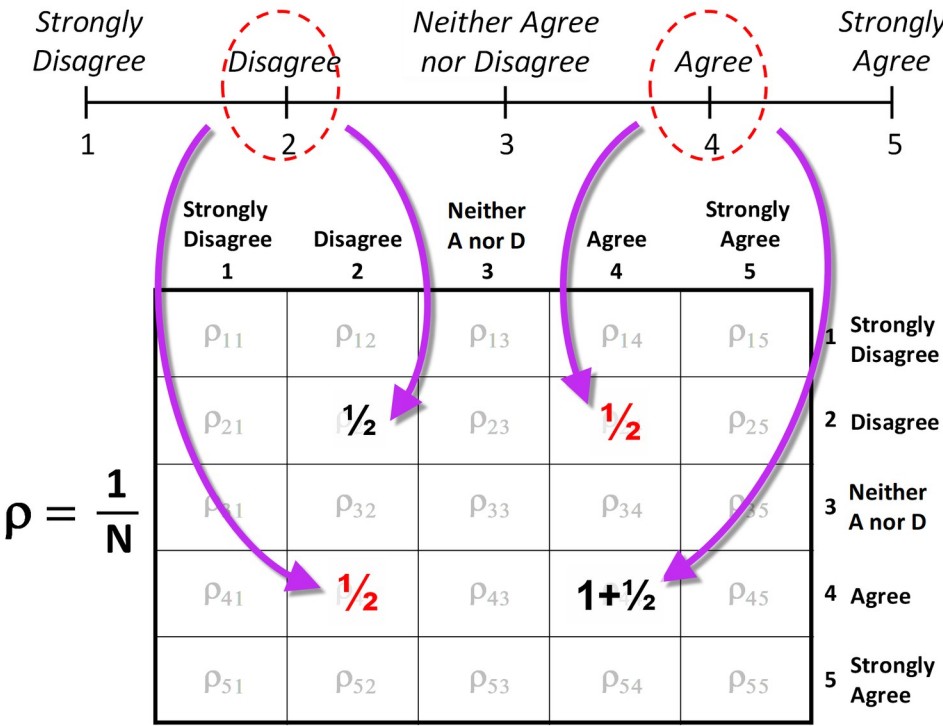

**Fig A2. For a double response, a half-count is added to each of the corresponding diagonal array elements, and a half-count to each corresponding off-diagonal element.**

box gives the probability that this individual strongly approves of Donald Trump's presidency. Alternatively, the sum of all off-diagonal boxes gives the probability that this participant is ambivalent with regard to Donald Trump's presidency (*i.e.*, the respondent agrees and disagrees with positive statements). Notice that summing the density matrices of all survey respondents, and then dividing by the number of respondents provides information on the population's approval/disapproval of Trump's presidency as well as the population's ambivalence towards Trump's presidency.

## Appendix B: Survey items from Study 1

Q ⇒ Quinnipiac Item,  BI ⇒ Brookings Institute Item

1. (BI) Trump's executive order [*Travel Ban*] is necessary to ensure the safety of all Americans. The government has a responsibility to ensure that nobody entering our country is coming here to do us harm. It is only reasonable to limit immigration from those six countries [*Iran, Somalia, Sudan, Yemen, Syria, and Libya*] until we know that the people coming do not pose a threat. While it may be true that many who do not pose a threat will be kept out in the process, our top priority must be reducing the threat to American citizens by keeping potential terrorists out.

2. (BI) The United States is a nation of immigrants, one with a long history of taking in those fleeing persecution in their home countries. Refugees have also contributed considerably to America. Data also shows that only a handful of refugees have been charged with terrorist plotting among hundreds of thousands who arrived to the United States in the past decade

and a half. The Constitution prohibits restricting immigration based on religion, which some courts have found this order [*President Trump's Travel Ban*] to do.

3. (BI) President Trump's Travel Ban is principally intended to keep Muslims out of the U.S.

4. (BI) I support President Trump's order of a missile strike against a Syrian air force base on April 6, 2017, after reports that the regime of Bashar Assad used chemical weapons that killed and wounded dozens of Syrians.

5. (BI) Thinking about how President Trump made his decision to launch a military attack on Syria, I have less confidence in the president's decision-making.

6. (Q) I approve of the way Donald Trump is handling his job as president.

7. (Q) My opinion of Donald Trump is unfavorable.

8. (Q) I would say that Donald Trump is honest.

9. (Q) I would say that Donald Trump does not have good leadership skills.

10. (Q) I would say that Donald Trump cares about average Americans.

11. (Q) I would say that Donald Trump is not level headed.

12. (Q) I would say that Donald Trump is a strong person.

13. (Q) I would say that Donald Trump is not intelligent.

14. (Q) I would say that Donald Trump is someone who shares my values.

15. (Q) I feel that Donald Trump is doing more to divide the country as President.

16. (Q) I think that Donald Trump is mostly bringing the right kind of change to the country.

17. (Q) I would say that Donald Trump is doing a worse job as president than I expected.

18. (Q) Having Donald Trump as President of the United States makes me feel safer.

19. (Q) I disapprove of the way the media has covered President Trump.

20. (Q) I approve of the way President Trump talks about the media.

21. (Q) I trust the news media more than Donald Trump to tell the truth about important issues.

22. (Q) I think that President Trump provides the United States with moral leadership.

23. (Q) I think that President Trump is abusing the powers of his office.

24. (Q) I approve of the way Donald Trump is handling race relations.

25. (Q) I would say that Donald Trump does not care about the issues facing minority groups in the United States.

26. (Q) As president, Donald Trump should continue tweeting from his personal Twitter account.

27. (Q) I think that President Trump's decisions and behavior as president have encouraged the white supremacist groups.

28. (Q) I approve of President Trump's response to the events in Charlottesville.

## Appendix C: Survey items from Study 2

F ⇒ Foreign Policy, E ⇒ Economy, I ⇒ Immigration, NM ⇒ News Media, MP ⇒ Mueller Probe, G ⇒ General

1. (G) I approve of the way Donald Trump is handling his job as president.

2. (E) In general, I do not think that the Trump administration is doing enough to help middle class Americans.

3. (NM) I am not concerned that President Trump's criticism of the news media will lead to violence against people who work in the news media.

4. (F) I disapprove of the way Donald Trump is handling foreign policy.

5. (I) In my opinion the Trump administration has not been aggressive enough in deporting immigrants who are here illegally.

6. (MP) I do not think that the FBI is biased against President Trump.

7. (E) I approve of the way Donald Trump is handling the economy.

8. (G) Donald Trump's presidency has made me think less favorably of the Republican party.

9. (F) I think that President Trump has strengthened the United States' position as the leader of the free world.

10. (NM) I approve of the way the news media has covered President Trump.

11. (MP) I think President Trump should fire Special Counsel Robert Mueller.

12. (I) I oppose the Trump administration's zero tolerance policy for undocumented immigrants, which means immigrants who illegally cross the border into the United States are immediately charged with a crime.

13. (G) In general, I like President Trump's policies.

14. (E) I would describe the state of the nation's economy as poor.

15. (NM) I approve of the way President Trump talks about the media.

16. (F) I disapprove of the way Donald Trump is handling the nation's policy towards Russia.

17. (I) I approve of the way Donald Trump is handling immigration issues.

18. (MP) I think President Trump should be impeached and removed from office if he fires Special Counsel Robert Mueller.

19. (E) I believe President Trump is more responsible for the current state of the economy than President Obama.

20. (G) I feel embarrassed to have Donald Trump as president.

21. (F) I have confidence in President Trump to handle the situation in North Korea.

22. (NM) I trust the news media to tell the truth about important issues more than President Trump.

23. (MP) I think that the investigation into any links or coordination between President Trump's 2016 election campaign and the Russian government is not a political witch hunt.

24. (G) I believe that President Trump wants to do what's best for himself.

25. (E) I think the nation's economy is getting better.

26. (F) I disapprove of the way Donald Trump is handling relations with European allies.

27. (I) I think the main motive behind President Trump's immigration policies is a sincere interest in controlling our borders.

28. (MP) I think that the investigation into any links or coordination between President Trump's 2016 election campaign and the Russian government is a legitimate investigation.

## Supporting information

**S1 Raw data.**
(XLSX)

## Acknowledgments

The authors thank Ingrid Morales, Leeann Ballejo, and Oliver Bineth for their invaluable assistance on many components of this project.

## Author Contributions

**Conceptualization:** James C. Camparo, Lorinda B. Camparo.

**Data curation:** James C. Camparo, Lorinda B. Camparo.

**Formal analysis:** James C. Camparo, Lorinda B. Camparo.

**Investigation:** James C. Camparo, Lorinda B. Camparo.

**Methodology:** James C. Camparo, Lorinda B. Camparo.

**Project administration:** Lorinda B. Camparo.

**Supervision:** Lorinda B. Camparo.

**Writing – original draft:** James C. Camparo, Lorinda B. Camparo.

**Writing – review & editing:** James C. Camparo, Lorinda B. Camparo.

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
