## [Decision Letter · Decision Letter 0]

4 Jan 2021

PONE-D-20-33843

Are Political-Opinion Pollsters Missing Ambivalence: “I love Trump”… “I hate Trump”

PLOS ONE

Dear Dr. Camparo,

Thank you for submitting your manuscript to PLOS ONE. After careful consideration, we feel that it has merit but does not fully meet PLOS ONE’s publication criteria as it currently stands. Therefore, we invite you to submit a revised version of the manuscript that addresses the points raised during the review process.

The referee finds the idea of ambivalence interesting, and suggested several ways to improve the paper. Also, I would argue that the concept of ambivalence across all political-party affiliations--with those defining themselves as independent showing the greatest degree of ambivalence--is a manifestation of the phenomenon of “voting contagion” observed in almost a century of presidential elections, see Braha, D., & De Aguiar, M. A. (2017). Voting contagion: Modeling and analysis of a century of US presidential elections. PloS one, 12(5), e0177970. It would be interesting if you could comment on this potential relationship.

We look forward to receiving your revised manuscript.

Kind regards,

Dan Braha

Academic Editor

PLOS ONE

Journal Requirements:

2. Please include captions for your Supporting Information files at the end of your manuscript, and update any in-text citations to match accordingly. Please see our Supporting Information guidelines for more information: http://journals.plos.org/plosone/s/supporting-information

Additional Editor Comments:

The referee finds the idea of ambivalence interesting, and suggested several ways to improve the paper. Also, I would argue that the concept of ambivalence across all political-party affiliations--with those defining themselves as independent showing the greatest degree of ambivalence--is a manifestation of the phenomenon of “voting contagion” observed in almost a century of presidential elections, see Braha, D., & De Aguiar, M. A. (2017). Voting contagion: Modeling and analysis of a century of US presidential elections. PloS one, 12(5), e0177970. It would be interesting if you could comment on this potential relationship.

Reviewers' comments:

Reviewer's Responses to Questions

**Comments to the Author**

1. Is the manuscript technically sound, and do the data support the conclusions?

Reviewer #1: Yes

2. Has the statistical analysis been performed appropriately and rigorously? 

Reviewer #1: Yes

3. Have the authors made all data underlying the findings in their manuscript fully available?

Reviewer #1: Yes

4. Is the manuscript presented in an intelligible fashion and written in standard English?

Reviewer #1: Yes

5. Review Comments to the Author

Reviewer #1: I really like this paper. I find the idea of ambivalence interesting and agree that it likely had more to do with the 2016 results than we previously understood.

There are a couple of things that I think would make the paper stronger.

1) Address whether it is necessarily ambivalence at play here. Hear me out here...I am a democrat and completely not ambivalent in my thoughts/opinions about Trump. However, some of the items on questionnaires would have me answering twice if given the option, not because of ambivalence, but because of different readings of the question. So, like, "the economy is doing well", I would think both "yes" because the stock market is awesome and "no" because, well, everything else. Or "Trump is strong", is yes in a bad way but no in a good way. So is it possible that you might be losing between party effect because of finding "ambivalence" where it doesn't really exist in democrats?

2) Why are you talking about both party and ideology and what is the basis for the hypothesizing about the relationship between lib/dem con/rep? There isn't much theoretical basis for worrying about both that I am aware of (and if there is, you should address it directly), so it seems to distract from your main topic. If you focused on either party (probably the stronger one for your purposes) or ideology, it would make your paper more concise and cogent. Get rid of the comparisons completely.

3) The literature on independent voters indicates that they are rarely truly independent with no leanings. It follows that their ambivalence might be predictable. It it probably worth mentioning this and citing the literature.

6. PLOS authors have the option to publish the peer review history of their article (what does this mean?). If published, this will include your full peer review and any attached files.

Reviewer #1: No

---

## [Author Response · Author response to Decision Letter 0]

29 Jan 2021

See the attached "Response to Reviewers" letter in the uploaded files.

---

## [Editor Report · Decision Letter 1]

10 Feb 2021

Are Political-Opinion Pollsters Missing Ambivalence: “I love Trump”… “I hate Trump”

PONE-D-20-33843R1

Dear Dr. Camparo,

We’re pleased to inform you that your manuscript has been judged scientifically suitable for publication and will be formally accepted for publication once it meets all outstanding technical requirements.

Kind regards,

Dan Braha

Academic Editor

PLOS ONE
---

## [Editor Report · Acceptance letter]

23 Feb 2021

PONE-D-20-33843R1 

Are Political-Opinion Pollsters Missing Ambivalence: “I love Trump”… “I hate Trump” 

Dear Dr. Camparo:

I'm pleased to inform you that your manuscript has been deemed suitable for publication in PLOS ONE. Congratulations! Your manuscript is now with our production department. 

Kind regards, 

on behalf of

Professor Dan Braha 

Academic Editor

PLOS ONE